# Adolescent screen time and unhealthy food consumption in the context of the digital development in New Caledonia

Akila Nedjar-Guerre[1]*, Guillaume Wattelez[1], Christophe Serra-Mallol[2], Stéphane Frayon[1], Olivier Galy[1]

1 Interdisciplinary Laboratory for Research in Education, EA 7483, University of New Caledonia, Nouméa, New Caledonia, 2 CERTOP, UMR 5044 CNRS, University of Toulouse Jean Jaurès, Toulouse, France

* akila.nedjar-guerre@unc.nc

**Data Availability Statement:** The data underlying the results presented in the study are available from Interdisciplinary Laboratory for Educational Research : dirlire@unc.nc Data cannot be shared

## Abstract

### Objective

For several years, the Pacific Island Countries and Territories (PICTs) have been undergoing digital developments that have had an impact on the time adolescents spend in front of screens. The association between screen time and the overconsumption of unhealthy foods has been observed but little studied in New Caledonia. The twofold aim of this research was to analyze adolescent screen time based on the number of screens at home, gender, place of living, ethnic community and family socio-professional category and determine the link with the consumption of unhealthy food and drinks.

### Methods

Self-report questionnaires on time spent in front of tablets, computers and mobile phones, as well as the consumption of unhealthy food and drinks, were administered between July 2018 and April 2019 to 867 adolescents from 11 to 15 years old during school hours in eight schools across the three provinces of New Caledonia.

### Results

Adolescents in rural areas had fewer screens than their urban counterparts, and the number of screens determined the amount of screen time, which was significantly higher among the adolescents living in urban areas (3.05 h/day weekdays vs rural 2.33 h/day). Screen time was not related to gender, socio-professional category or ethnic community, but correlations were found between screen time and consumption of unhealthy food and drinks. Those who consumed less than 1 u/day of unhealthy drinks watched screens for 3.30 h/day, whereas those who consumed more than 1 u/day watched screens for 4.13 h/day. Also, those who consumed less than 1 u/day of unhealthy food watched screens for 2.82 h/day and those consuming more than 1 u/day did so for 3.62 h/day. Melanesians and Polynesians consumed greater quantities of unhealthy food and drinks than Europeans. As the consumption of unhealthy products is linked to screen time in the context of digital development, there is

publicly but The data underlying the results presented in the study are available on demand from Interdisciplinary Laboratory for Educational Research : dirlire@unc.nc 1. GDPR (General Data Protection Regulation). According to GDPR, this research used sensitive data. This is mainly minors (under 16) who are considered "vulnerable persons". In addition, we used ethnicity which is classified as sensitive personal data. 2. Statistical confidentiality principles. We worked with non-identifying data for this research, but we cannot assume that this data is fully anonymised. Our small sample size in some categories means that individualisation is possible with our dataset.

**Funding:** The author(s) received funding from Fondation Nestlé and from University of Caledonia for this work. The funders had no role in study design, data collection and analysis, decision to publish, or preparation of the manuscript. The authors did not receive a salary from the funders.

**Competing interests:** The authors have declared that no competing interests exist.

an urgent need to address the excessive consumption of unhealthy foods in Oceanian populations, particularly among young people.

## Background

As in the rest of Oceania during the nineteenth century, New Caledonia underwent a socioeconomic transition that changed dietary practices. Western colonization, with the mass industrialization of food, spread cheap and low-cost products among the indigenous populations, beginning first with the French troops on the move, then among the urban and proletarian populations. New Caledonians thus rapidly integrated tinned food, sweetened condensed milk, cakes and pulses into their daily diet, and the working population became increasingly monetarized as local commerce was established [1]. The Second World War had a significant impact on New Caledonia with the installation of American bases, which increased communications and trade, particularly in food, and generalized a monetary economy. Urban migration was accompanied by changes in agricultural techniques and methods and food practices, with traditional farming, fishing and hunting activities abandoned in favor of paid work [2] in mines or in the service sector. From self-production and exchange activities, the inhabitants moved to the monetary acquisition of their food, in particular meat and fish (canned), flour and rice, which supplemented and then gradually replaced local plants, the surface area of which was reduced in favor of cash crops and livestock farming.

In the 1970s, the first supermarkets opened, first in Nouméa agglomeration and then in the Northern Province villages. Hypermarkets followed in the 1980s-1990s, with a considerable opening up of the range of food choices through industrial and marketing diversification. However, the growing weight of social and income inequalities in access to monetized food has materially restricted this opening up, leaving the most heavily industrialized products to the poorest. This situation has occurred even though inclusion in exchange networks seems to be inversely proportional to the use of the market, which is even stronger the further one moves away from Nouméa and the monetization of food exchanges [3], as has also been demonstrated in French Polynesia [4]. With easier access to processed food, the place of junk food and sugar-sweetened beverages has grown, leading to unhealthy dietary habits.

At the same time, New Caledonia and all of Oceania have been undergoing striking developments in digital access over the last thirty years. Digital expansion has been one of the major transitions in human development around the world [5]. The global Internet and its social networking platforms have become the primary communication system and these all-purpose technologies are clearly transforming processes and lifestyles in all domains. In the Pacific Island Countries and Territories (PICTs), where access to digital technologies is fairly recent and screen use is becoming democratized [6], strong digital development is underway. Most PICTs increased to 3G and even 4G coverage between 2010 and 2016, and the number of people connected to the Internet doubled and even tripled in some countries [7]. The percentage of households with an Internet connection increased between 2010 and 2016 for several island territories. For example, it increased from 18.8% to 33.6% in Fiji, from 5.4% to 29.5% in Vanuatu, from 2.9% to 8.5% in the Solomon Islands, and from 2.2% to 8.8% in Papua New Guinea, thus indicating disparities between these PICTs [7]. In New Caledonia, the number of connected screens in households has increased and the digital network has steadily improved over in the past several years [8]. The Government launched a strategic plan to deploy digital technologies and offer equal access to the digital environment to the three provinces of New

Caledonia [9]. Through the Gondwana 1 communication cable coming from Australia since 2008, New Caledonia has gained increasingly more access to the Internet and more modern telecommunication services, in major part due to strategic territorial planning. This national strategy increased the rate of Internet users among young people from 30% in 2011 to 93% in 2018 [8]. Fig 1 shows the distribution of inhabitants over the territory and the concentration of urban areas in the Nouméa agglomeration. Connectivity has increased throughout the territory, including in rural areas as shown in Fig 2A–2C.

New Caledonia is an ideal PICT for studying the complexities of the socioeconomic and digital developments in the Pacific because of its cultural diversity and gradients of urbanization and digital infrastructure development. Except for Nouméa agglomeration, the major part of New Caledonia is considered rural with low population density (Fig 1). Digital infrastructures are highly developed in Southern Province with its main city, Nouméa, and the neighboring towns, whereas it is less developed in Northern Province and Loyalty Islands Province, which are mainly rural [8] with traditional tribal populations. The towns involved in our study show differences in Internet connection (Fig 2A), computer equipment (Fig 2B) and mobile phones (Fig 2C) between the towns in Southern Province (Nouméa, Païta, Dumbéa), Northern Province (Koumac, Poindimié) and Loyalty Islands Province (Lifou), although connectivity showed a general increase across the whole territory between 2009 and 2019.

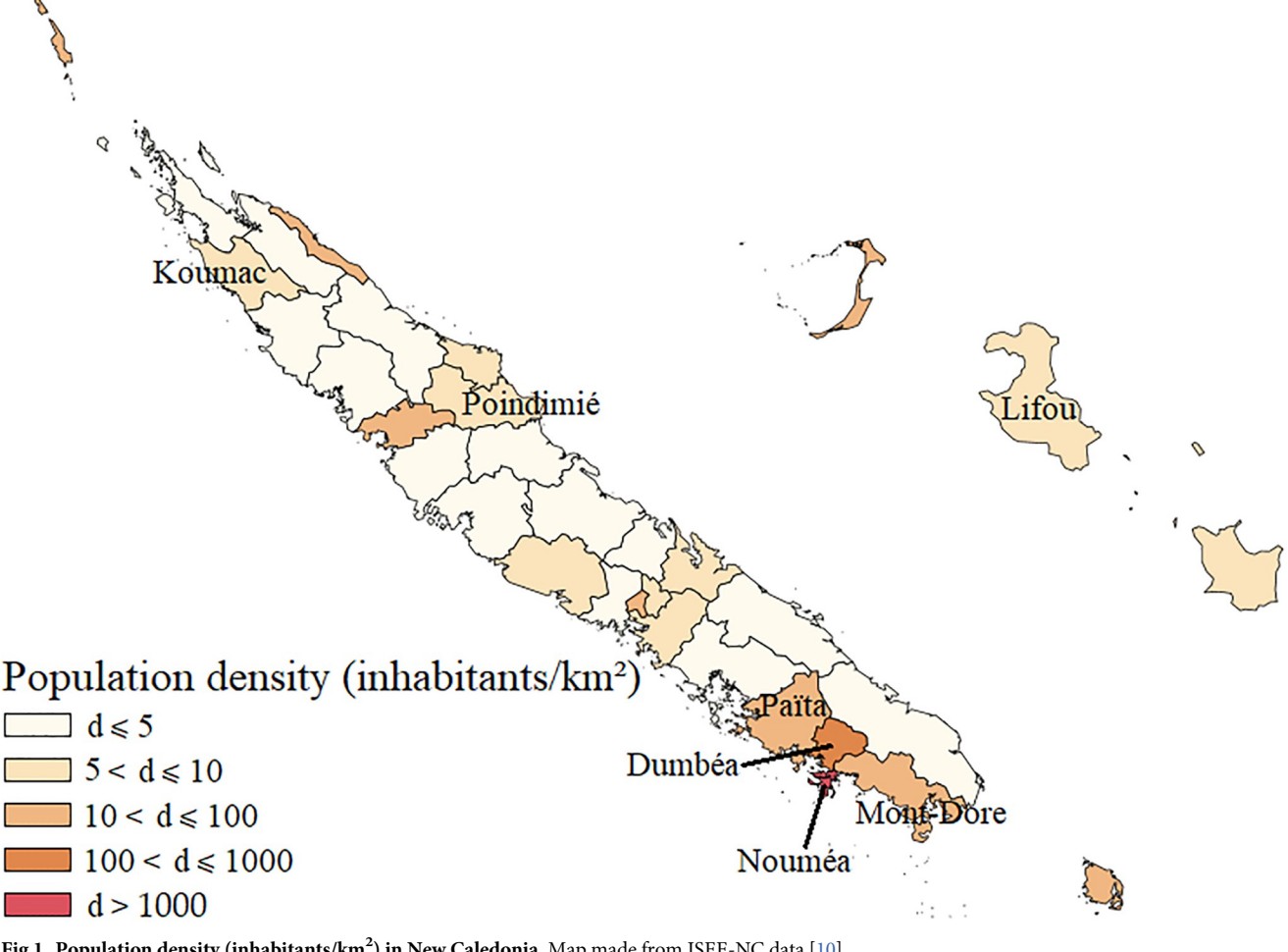

**Fig 1. Population density (inhabitants/km$^2$) in New Caledonia.** Map made from ISEE-NC data [10].

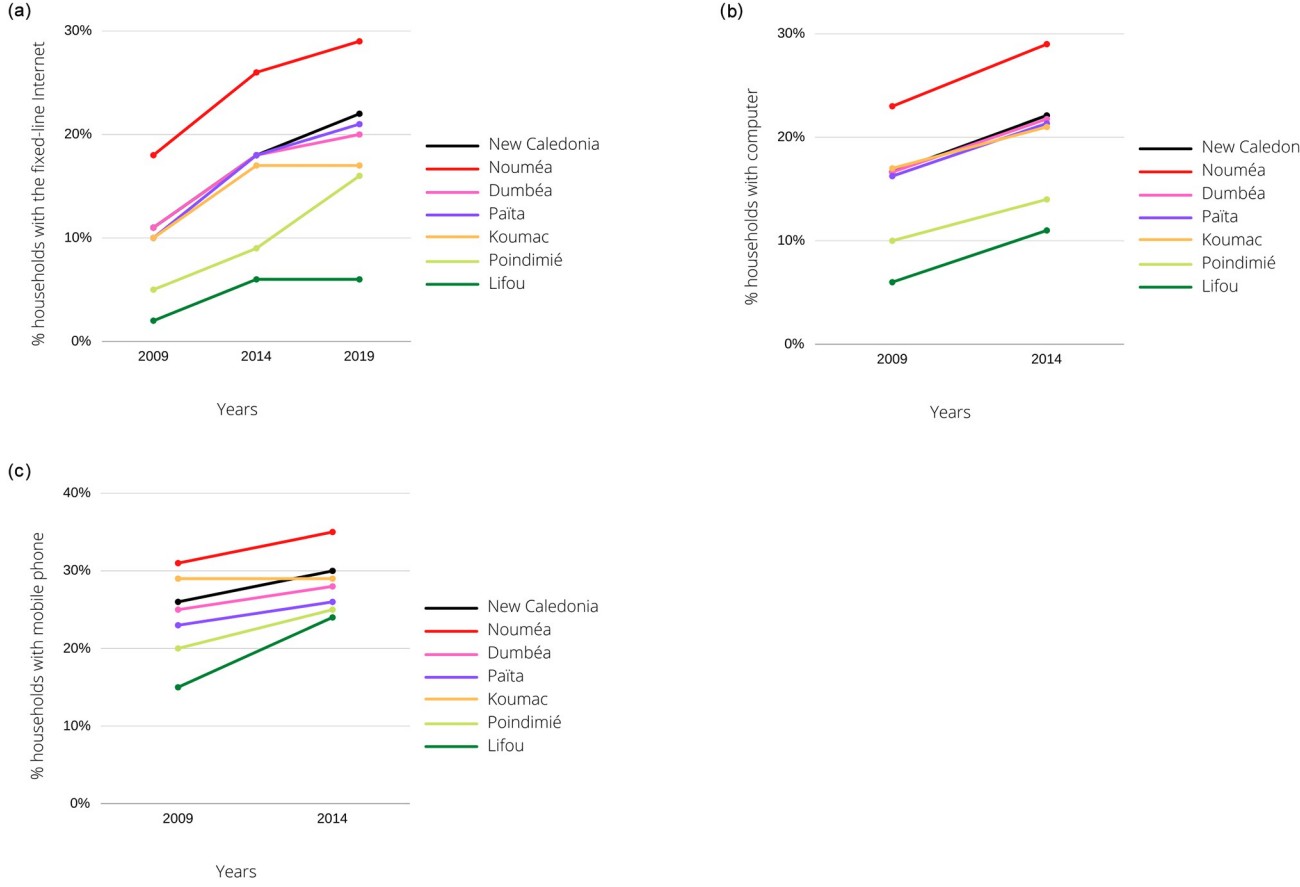

**Fig 2.** A. Evolution of the fixed-line Internet connection between 2009 and 2019 in New Caledonia households (Nouméa, Dumbéa, Païta, Koumac, Poindimié, Lifou). Data from ISEE-NC [10]. B. Evolution of computer ownership between 2009 and 2014 in New Caledonia households (Nouméa, Dumbéa, Païta, Koumac, Poindimié, Lifou). Data from ISEE-NC [10]. C. Evolution of mobile phone ownership between 2009 and 2014 in New Caledonia households (Nouméa, Dumbéa, Païta, Koumac, Poindimié, Lifou). Data from ISEE-NC [10].

Young New Caledonians have increasingly turned to screen use, with mobile phones, tablets and computers, instead of traditional media (television and print media). Indeed, in 2018 [8], 93% of young people (from 18 to 25 years old) were surfing on the Internet for an average of 4 hours 11 minutes per day, whereas in France the Internet was used 5 hours 48 minutes per day by 18- to 34-year-olds in 2019 [11]. In 2011, one in five mobile phones was a smartphone owned by 16% of the population over 15 years old. Smartphone use increases with age, level of education and socio-professional category. This digital development is also transforming the modes of screen use: from single-family consumption of traditional television screens (one screen for several people) to more personal consumption with the acquisition and development of individual mobile phones [12].

The digital development of territories is often unequal, depending on the type of territory (i.e., urban or rural), which may explain some of the differences in screen use by the inhabitants: for territories undergoing digital development, the literature shows that people living in rural areas are less likely to have access to the Internet and equipment than in urban areas [13]. The different rate of Internet access among telephone subscribers might be explained by differences in infrastructures across the territory depending on the place of living (i.e., rural or urban): in 2011, this rate varied from 14% to 76% and in 2018, 83% of the population was connected to the Internet in urban areas and 74% in rural areas [8].

Similarly, the socioeconomic resources of families [10] may also explain the time young people spend in front of screens, since families with low socioeconomic status are less equipped with digital technologies [14]. Yet surprisingly, studies have found that children from families with low incomes might be more likely to have high screen time [15]. Indeed, children in higher income families might have lower media consumption because of parental insistence on restricting screen use [16]. Thus, screen time may depend on access to equipment, lifestyle and place of living: in rural areas, most youths spend time outdoors, and many of them are more connected to nature compared to urban youths, who are more connected to screens [17]. While New Caledonian adolescents living in rural areas are mainly Melanesian [10], little is known about the relationship between screen time and place of living or ethnicity. In contrast, ethnicity was shown to be a factor influencing screen time in New Zealand [18], indicating that indigenous people spent more time in front of screens. Melanesian people of New Caledonia, who mostly live in tribes in rural areas, spend long hours outdoors [19], with greater outdoor socialization and higher physical activity among adolescents, particularly boys [20]. In tribes, the daily activities of young people are related to the type of social organization of the group into clans and to the needs for economic adaptation. The productive activities useful to the community are numerous: working in the fields, collecting natural products, taking care of animals, looking after babies (for girls), and "household" activities such as raking around the hut. Similarly, social exchanges and interactions between clan members are frequent and take priority over individual actions [19]. In contrast, Melanesians living in urban areas have a lifestyle almost similar to that of other communities.

As a consequence, these discrepancies between communities and places of living [3, 21] may have a strong influence on screen use.

Owning and using screens increase physical inactivity, as people tend to spend considerable time in front of their screens [22]. The relationship between the increase in the number of home devices (television, tablet, telephone, video games, etc.) and a given population's obesity rate has been widely observed and can take four forms [23]: (1) the increase in immediate food intake during screen time [24], (2) the unconscious and diffuse effects of advertising for products of poor nutritional quality [25], (3) sedentary lifestyles [26], and [4] the effect of screen use in reducing sleep time, a risk factor for obesity [27]. These factors, when combined, can lead to health issues, especially in countries undergoing digital development. A previous study involving Melanesian adults living in Vanuatu showed that owning electronic devices (mobile phones, music players, television, game consoles) was positively correlated with obesity [28]. Positive relationships have also been shown between screen time and the consumption of unhealthy foods and drinks in Japanese adolescents [29]. Although studies have been conducted in PICTs to examine the use of digital devices and its links with overweight and obesity [28], the literature is very sparse regarding these links in Pacific Islander adolescents according to ethnic communities, and the influence of weekday and weekend screen use on their food behaviors remains unclear.

While little is known about the impact of the New Caledonian digital development on adolescent health, specifically on unhealthy diets, it has been clearly established that 35% of the 11- to 16-year-olds are now overweight or obese [21]. Diet is one of the causes, given the high intake of both unhealthy beverages and foods of poor nutritional quality among these adolescents [30]. The place of living and the ethnic community are also involved: adolescents living in rural areas consume significantly more unhealthy foods and drinks than those in urban areas [21, 31, 32]. We might also suspect that screen use has an influence on individual behaviors, with a significant impact on adolescent diets and health outcomes. We therefore hypothesized that high levels of screen time for New Caledonian adolescents would be linked to the consumption of unhealthy food and drinks.

This study examined home digital equipment and the weekday and weekend screen time distribution and the links between screen time and the consumption of unhealthy food and drink in New Caledonian adolescents according to their place of living, ethnicity, gender and socioeconomic variables.

## Methodology

In this study, quantitative data were obtained from questionnaire responses. Our study was part of a community-based food culture project conducted in eight selected representative school sites in the three provinces (Northern Province, Southern Province and Loyalty Islands) of New Caledonia. The schools were selected on the basis of the following criteria: (1) a representative repartition of the schools between rural and urban areas (respectively, 37% and 63% of the population) and (2) sufficient school size ($n > 200$). Selected schools were then randomly drawn from among these eligible schools: five in Southern Province (urban), two in Northern Province (rural) and one in Loyalty Islands Province (rural).

Ethics approval for this project was obtained from the Consultative Ethics Committee of New Caledonia (CCE 2018–06 001) and the project was conducted in accordance with the requirements of the Declaration of Helsinki.

Data were collected before the Covid-19 epidemic, between July 2018 and April 2019, from 1060 school-going adolescents (11–15 years old) during class time. In each school, two classes per level (6th, 5th, 4th, 3rd) were chosen to respond to the anonymous questionnaire, which consisted of two parts lasting 30 minutes each and was carried out in two stages. In each school, 95% of the expected participants responded due to absences or parental refusal. We removed questionnaires with incomplete responses and obtained 867 completed questionnaires with parental permission. Written consent was obtained from the children's parents before the start of the study. The project was first authorized by the Vice-Rectorate and then presented to the school directors. The targeted schools were contacted, and the project was then proposed to the teaching teams for their acceptance.

The students answered online via digital tablets or, if necessary, directly on computer stations in classrooms. The language used in this questionnaire was French, the official language understood by the adolescents and used for teaching. Concerning media use, questions were asked about the number of devices with digital screens in their families (e.g., number of computers, tablets and mobile phones and connected fixed screens) and the time they spend on them during the week and on weekends. Screen time was assessed through the following two questions: *"How much time do you spend in front of the computer or tablet during the week*? and *"If you have a mobile phone, how much time are you logged in on your phone during the week*?". The answers to these two questions about screen time per day during the school week (Monday to Friday) were then added together. The same questions were also asked about the weekends and then these two factors (weekday and weekend use) were combined to give the total screen time for the whole week as follows: average time = 5 x time (weekdays) + 2 x time (weekend) divided by seven.

Ethnicity was self-reported by the adolescents and classified according to the recommendations of the INSERM report on New Caledonia [33], but we did not allow the students to report belonging to more than one ethnic group. Ethnic categorization was based on the question: "*Which community do you feel you belong to*?" We proposed a category called "other" in which we grouped together all ethnic groups that represented less than 5%, such as Chinese and African.

Socioeconomic status (SES) was indexed on the basis of the occupation of the household reference person (defined as the head of the household with the highest income) using the

socioeconomic classification of national statistics [34]. In this study, three categories were selected: management and professional occupations (upper SES), intermediate occupations (intermediate SES), and common and manual occupations (lower SES).

On the basis of the last census in New Caledonia [10], the area of residence was determined using a European standard [35]. Densely populated areas comprising at least 50,000 inhabitants in a continuous zone with more than 500 inhabitants/km$^2$ were classified as urban. Areas with more than 50,000 inhabitants in a continuous area of over 100 inhabitants/km$^2$ or areas adjacent to an urban area were classified as semiurban. Rural areas were defined as those areas that did not fulfill the conditions required to qualify as urban or semiurban (Fig 1). We did not distinguish between urban and peri-urban in our field analysis.

To assess diet, a food frequency questionnaire (FFQ) consisting of 28 questions was set up to obtain data on the frequency of food consumption [31], but we used only the responses regarding unhealthy food and drinks: foods high in salt, sugar or saturated fat (french fries, processed salted meats, chocolate and confectionery, cakes, pastries and cookies) and beverages that are not recommended. We complied with the Pacific Community (SPC) standards for unhealthy products, which advise a maximum of one unit per day [30]. In addition to food and drink consumption, additional questions on food purchases on the way to and from school and at the school canteen were included in the questionnaire. The intake frequencies for the individual food items of the FFQ were first converted to daily frequency equivalents (DFEs), calculated by allocating proportional values to the original frequency categories with reference to a base value of 1.0, equivalent to once a day.

## Statistical analysis

First, the significant differences according to place of living (rural vs urban) were tested by using means equality tests (Student or Welch's t-test) for numerical variables. For categorical variables, the $\chi^2$ test was used when Cochran's rule was satisfied, otherwise Fisher's exact test was used.

The differences in screen time according to the number of electronic devices available at home were tested with the Kruskal-Wallis test because the ANOVA hypotheses were not satisfied. When a difference was significant, the Steel-Dwass-Critchlow-Fligner post-hoc test was implemented to determine which ones were different from each other.

Second, a multilinear regression was computed to determine the factors associated with the adolescents' screen times. As no significant interaction was found between factors, we used a type 2 sum of squares for ANOVA. Last, we performed logistic regressions to highlight associations between screen time and consumption of unhealthy food and drinks.

The analyses were conducted with R 3.5.1 [36] with a first species risk set at $\alpha = 0.05$. Graphs were built with ggplot2 package [37].

## Results

Table 1 presents the overall descriptive data according to place of living (i.e., rural vs urban).

In this sample, rural areas had more Melanesians (75%) and families with low SES (74.38%), and the difference was significant between the adolescents in the rural and urban areas for the consumption of unhealthy food (3.90 units compared to 3.27 units in urban areas, $p < 0.001$) and unhealthy drinks (1.08 units compared to 0.91 units in urban areas, $p = 0.024$). Moreover, the difference was also significant between rural (36.73%) and urban (27.44%) areas for the number of adolescents exceeding the SPC standards for unhealthy drinks. No significant difference was observed between groups concerning unhealthy food

**Table 1. Sociodemographic characteristics, unhealthy food and drinks consumed by adolescents, time spent on electronic screen devices, and number of screens according to place of living.**

| | | Rural | Urban | p-value | Total |
|---|---|---|---|---|---|
| | | n (%) | n (%) | | n (%) |
| Whole sample | | 324 (37.37) | 543 (62.63) | | 867 (100) |
| Gender | Girls | 170 (52.47) | 257 (47.33) | 0.659 | 427 (49.25) |
| | Boys | 154 (47.53) | 286 (52.67) | | 440 (50.75) |
| Ethnicity | Melanesian | 243 (75.00) | 154 (28.36) | < 0.001 | 397 (45.79) |
| | European | 69 (21.30) | 257 (47.33) | | 326 (37.60) |
| | Polynesian | 6 (1.85) | 83 (15.29) | | 89 (10.27) |
| | Other | 6 (1.85) | 49 (9.02) | | 55 (6.34) |
| SES | Upper | 51 (15.74) | 198 (36.46) | < 0.001 | 249 (28.72) |
| | Intermediate | 32 (9.88) | 84 (15.47) | | 116 (13.38) |
| | Lower | 241 (74.38) | 261 (48.07) | | 502 (57.90) |
| Unhealthy food | ≤ 1 u/day | 20 (6.17) | 50 (9.21) | 0.145 | 70 (8.07) |
| | > 1 u/day | 304 (93.83) | 493 (90.79) | | 797 (91.92) |
| Sweet beverages | ≤ 1 u/day | 205 (63.27) | 394 (72.56) | 0.005 | 599 (69.09) |
| | > 1 u/day | 119 (36.73) | 149 (27.44) | | 268 (30.91) |
| | | mean ± sd | mean ± sd | p-value | mean ± sd |
| Nutrition (unit/day) | Unhealthy food | 3.90 ± 2.50 | 3.27 ± 2.26 | < 0.001 | 3.50 ± 2.37 |
| | Unhealthy drinks | 1.08 ± 1.17 | 0.91 ± 1.12 | 0.024 | 0.97 ± 1.14 |
| Screen time (h/day) | Computer and tablets weekdays | 1.27 ± 0.99 | 1.77 ± 1.19 | < 0.001 | 1.58 ± 1.15 |
| | Computers and tablets weekends | 2.65 ± 1.81 | 3.10 ± 1.81 | < 0.001 | 2.93 ± 1.82 |
| | Phone weekdays | 1.06 ± 0.82 | 1.28 ± 0.88 | < 0.001 | 1.20 ± 0.86 |
| | Phone weekends | 2.30 ± 1.91 | 2.72 ± 2.01 | 0.003 | 2.56 ± 1.98 |
| Number of screens | None | 11 (3.40) | 5 (0.92) | < 0.001 | 16 (1.85) |
| | One | 46 (14.20) | 26 (4.79) | | 72 (8.30) |
| | Two | 80 (24.69) | 89 (16.39) | | 169 (19.49) |
| | Three or more | 187 (57.72) | 423 (77.90) | | 610 (70.36) |

Numbers indicate size and percentage "n (%)" for the categorical variables (gender, ethnicity, SES, unhealthy food, sweet beverages and number of screens) and "mean ± standard deviation" for the numerical variables (consumption of unhealthy food and drink and screen time).

consumption, but a significant difference was observed between rural adolescents (1.08 u/day) and urban adolescents (0.91 u/day) for unhealthy drink consumption.

Screen time was lower in rural areas (2.33 h/day in rural vs 3.05 h/day in urban areas), and this difference was significant for all types of screens (computers, tablets and mobile phones) and for both weekdays and weekends.

With regard to digital equipment, significantly more adolescents living in urban areas had more than three screens at home (77.9%) compared to rural adolescents (57.72%), and families living in urban areas had significantly more screens (p < 0.001).

Both groups showed significant differences in screen time on weekdays and weekends: screen time was doubled on weekends compared to weekdays (2.78 h/day on weekdays vs 5.49 h/day on weekends) for computers, tablets and phones.

Table 2 shows the results of the multilinear regressions to explain screen time according to socioeconomic factors and number of screens.

Screen time was positively correlated with the place of living (p < 0.001), the "other" community (p = 0.018), and the number of screens (p = 0.014 for one screen, p = 0.002 for two screens and p = 0.001 for three screens or more). Screen time was not related to socio-

**Table 2.  Screen time according to associated factors (gender, place of living, ethnicity, socio-professional status and number of screens in the home).** Description and multiple linear regression.

| | | Screen time (h/week) | | Linear regression | |
| | | Mean ± SD | p-value | B [95% CI] | p-value |
|---|---|---|---|---|---|
| Gender | Girls | 3.66 ± 2.07 | 0.133 | | |
| | Boys | 3.46 ± 1.99 | | -0.21 [-0.47;0.05] | 0.107 |
| Place of living | Rural | 3.08 ± 1.87 | < 0.001 | | |
| | Urban | 3.84 ± 2.07 | | 0.37 [0.07;0.68] | 0.018 |
| Ethnic community | European | 3.18 ± 1.91 | < 0.001 | | |
| | Melanesian | 3.86 ± 2.07 | | -0.11 [-0.44;0.23] | 0.531 |
| | Polynesian | 3.50± 1.94 | | 0.25 [-0.22;0.73] | 0.294 |
| | Other | 4.16 ± 2.16 | | 0.76 [0.2;1.31] | 0.008 |
| SES | Upper | 3.86 ± 2.07 | 0.002 | | |
| | Intermediate | 3.80 ± 2.01 | | 0.11 [-0.32;0.54] | 0.618 |
| | Lower | 3.35 ± 1.99 | | -0.17 [-0.51;0.16] | 0.311 |
| Number of screens | None | 1.27 ± 0.52 | < 0.001 | | |
| | One | 2.65 ± 1.82 | | 1.32 [0.27;2.38] | 0.014 |
| | Two | 2.99 ± 1.64 | | 1.58 [0.59;2.58] | 0.002 |
| | Three or more | 3.88 ± 2.07 | | 2.32 [1.34;3.29] | < 0.001 |

professional status or gender. Similarly, there was no significant difference in screen time between the European, Melanesian and Polynesian communities.

Table 3 shows that the consumption of unhealthy drinks was significantly correlated with the ethnic community: the European adolescents were less likely to consume more than 1 u/day of unhealthy drinks than the Melanesians ($OR_a = 3.07$, p < 0.001) and the Polynesians ($OR_a = 2.03$, p = 0.015).

A positive association was observed between screen time and consumption of unhealthy drinks (OR = 1.44, p < 0.001). Beyond the recommended standard of drinks to be consumed per day (> 1 u/day), screen time was significantly higher (3.30 h/day vs 4.13 h/day).

**Table 3.  Consumption of unhealthy drinks and unhealthy food according to the factors under study (gender, place of living, ethnicity, socio-professional status and screen time).** Description and adjusted odds ratios.

| | | Unhealthy drinks | | Logistic regression | | Unhealthy food | | Logistic regression | |
| | | ≤ 1 u/day | > 1 u/day | $OR_a$ [95% CI] | p-value | ≤ 1 u/day | > 1 u/day | $OR_a$ [95% CI] | p-value |
|---|---|---|---|---|---|---|---|---|---|
| Gender [n (%)] | Girls | 303 (70.96) | 124 (29.04) | | | 37 (8.67) | 390 (91.33) | | |
| | Boys | 296 (67.27) | 144 (32.73) | 0.9 [0.55;1.45] | 0.660 | 33 (7.50) | 407 (92.50) | 1.29 [0.78;2.14] | 0.322 |
| Place of living [n (%)] | Rural | 205 (63.27) | 119 (36.73) | | | 20 (6.17) | 304 (93.83) | | |
| | Urban | 394 (72.56) | 149 (27.44) | 0.87 [0.4;1.9] | 0.728 | 50 (9.21) | 493 (90.79) | 2.26 [0.66;8.02] | 0.201 |
| Ethnic community [n (%)] | European | 266 (81.60) | 60 (18.40) | | | 35 (10.74) | 291 (89.26) | | |
| | Melanesian | 237 (59.70) | 160 (40.30) | 3.07 [2.04;4.66] | < 0.001 | 22 (5.54) | 375 (94.46) | 1.28 [0.38;4.19] | 0.691 |
| | Polynesian | 60 (67.42) | 29 (32.58) | 2.03 [1.14;3.56] | 0.015 | 10 (11.24) | 79 (88.76) | 0.67 [0.16;3.02] | 0.585 |
| | Other | 36 (65.45) | 19 (34.55) | 2.04 [1.05;3.9] | 0.032 | 3 (5.45) | 52 (94.55) | 102.45 [1.78;635432.10] | 0.115 |
| SES [n (%)] | Upper | 193 (77.51) | 56 (22.49) | | | 24 (9.64) | 225 (90.36) | | |
| | Intermediate | 86 (74.14) | 30 (25.86) | 0.92 [0.53;1.58] | 0.757 | 9 (7.76) | 107 (92.24) | 0.96 [0.43;2.31] | 0.927 |
| | Lower | 320 (63.75) | 182 (36.25) | 1.34 [0.89;2.03] | 0.158 | 37 (7.37) | 465 (92.63) | 1.02 [0.54;1.89] | 0.960 |
| Screen time [h/day] | Mean ± SD | 3.30 ± 1.95 | 4.13 ± 2.10 | 1.44 [1.26;1.65] | < 0.001 | 2.82 ± 1.99 | 3.62 ± 2.02 | 1.94 [1.22;3.39] | 0.011 |

Consuming more than 1 u/day of unhealthy food was significantly correlated with screen time ($OR_a$ = 1.94, p = 0.011). Unhealthy food consumption was not correlated with gender, place of living, ethnic community or SES.

## Discussion

This study showed considerable disparity in the number of digital devices per household during this period of digital development in New Caledonia. Notably, our study revealed differences between places of living (i.e., rural vs urban). We found a significantly lower number of screens in rural families compared to their urban counterparts (57.72% in rural areas vs 77.90% in urban areas owned three devices or more). At the same time, interesting correlations were observed between screen time and the consumption of unhealthy foods in the 11- to 15-year-old adolescents. We showed that the consumption of unhealthy food and drinks was positively associated with screen time in these adolescents, whose screen time per day was higher in urban areas (3.05 h/day on weekdays and 4.95 h/day on weekends) than in rural areas (2.33 h/day on weekdays and 5.82 h/day on weekends).

### Digital equipment at home and screen use in rural and urban areas

Digital development in New Caledonia has been different in rural areas and urban areas. Rural households own significantly fewer screen devices compared to those in urban areas. Indeed, more screens resulted in higher screen times for these New Caledonian adolescents (Table 1). Although Internet connections have steadily increased in recent years, with 82% of the territory now covered by 4G [8], ownership of computers and mobile phones was significantly different between the rural and urban areas. Many homes in rural areas had no screens or only one screen: 42.29% of young New Caledonians in rural areas had two screens or less, with television included in the number of screens, compared to 22.10% in urban areas. The private and intimate spaces such as bedrooms, which are generally reserved for personal mobile use [12], are less in keeping with New Caledonian tribal housing, which has traditionally been huts [38]. Urban households tend to have more of these private and intimate spaces, facilitating the more personal use of screens. Indeed, screen use as a leisure activity doubled at weekends compared to weekdays in both rural and urban areas. The digital development is changing leisure practices, leaving less time for physical activities [39] as screen times increase. The daily screen time of adolescents was compared with that of adolescents in a similar educational system (with similar school days, school programs and activities) in Metropolitan France, showing lower values for the New Caledonian adolescents in every socioeconomic category, with 57% of French adolescents spending at least 4 h/day in front of screens and 39% at least 6 h/day [40]. In 2016, 72% of the children in secondary school in Australia [41] spent 2 hours or more per weekday in small screen entertainment (including television, computers, video and DVDs). Given the increase in digital equipment combined with the successive lockdowns for the Covid-19 pandemic, it seems likely that the screen times of New Caledonian adolescents will soon reach levels comparable to those of Metropolitan French and Australian adolescents.

### Screen time and associated factors

Place of living influenced the screen time of the New Caledonian adolescents, but it should be noted that the rural lifestyle, with much more time spent outside [19], and the lower number of devices in rural compared to urban areas, likely explained the lower use of screens. Tribal life is highly developed in rural areas and external interactions in the community are important. Individuals develop networks of economic and community sociability in both rural and urban areas, particularly in informal settlements. Although the Melanesian community is

much larger in rural areas [10], the difference in screen time depended on their place of living (rural or urban) and not ethnicity, as we observed no link between Melanesian and European adolescents or between Polynesian and European adolescents in screen time.

Urbanization is also leading to a decrease in the number of rural settlements and an increase in private spaces in homes, which may explain the higher screen time with more mobile devices in urban families.

As Table 2 shows, the SES of the parents was not a factor that explained the differences in screen time, but digital devices were more prevalent in urban areas, where more families have higher SES.

Similarly, there was no difference between gender category and screen time among New Caledonians in the 11-to-15-year-old age category. Although the differences in screen time were not significant between girls and boys, the types of activities carried out on the Internet might differ, with mainly screen-based games for boys and social networking for girls [42], but our data did not allow us to make such observations.

## Unhealthy food and drinks and associated factors

New Caledonian adolescents who had higher screen times tended to have a higher consumption of unhealthy food and drinks, as defined in the SPC recommendations [30] (Table 3). This supports the observation that adolescents tend to consume unhealthy food and drinks during their screen time [24]. This has already been reported in other multicultural countries: young people consume unhealthy food in front of screens in Brazil [43] and Colombia [44], where this link between unhealthy food consumption and screen time is greater in urban areas. Indeed, media advertising may encourage adolescents to consume more unhealthy food and drinks [25, 29, 45]. Also, although the strength of the association between screen use and unhealthy food consumption may depend on the type of screen, we did not further investigate this link [46].

Screen time was lower for the rural adolescents than for their urban counterparts (Table 1), but their consumption of unhealthy drinks was generally higher, particularly among Melanesian adolescents compared to European adolescents (p < 0.001 for Melanesians and p = 0.015 for Polynesians), and this might increase with screen time (Table 3). Among Polynesians, whose representation of eating well corresponds to eating a lot [47], the consumption of unhealthy food was greater than among Melanesians. The consumption of unhealthy drinks by Melanesians [3] exceeded the SPC norms (40.3% drank more than 1 u/day vs 18.4% in European adolescents). The notion of "eating a lot" as "eating well" is often cited by low-income families in urban areas, as well as by traditional Melanesian families in rural and tribal areas [3]. This difference between Melanesians and Europeans in unhealthy food and drink consumption can also be explained by family regulation and parental control of adolescents. The role and place of families in health education differ and are probably less important in Melanesian than in European populations [3, 48], which may contribute to the dietary balance of adolescents. This might explain why the Melanesians and Polynesians consumed higher quantities of unhealthy drinks in our study (Table 3).

The parents of the Melanesian and Polynesian adolescents probably had little health literacy [49] with regard to assessing the nutritional qualities of the unhealthy drinks that their adolescents consumed in greater quantities than did the other ethnic groups. The lack of information no doubt lowered their capacity to act and process information appropriately [48]. Knowledge about foods and their "benefits" is not necessarily conveyed, particularly within Melanesian and Polynesian families, as has been noted among Samoan adults [50], and this might also explain the overconsumption of unhealthy drinks by Melanesian adolescents.

In New Caledonia, digital development is well advanced and the country ranks third in the development of connectivity among Oceanian countries, after Australia and New Zealand [8]. In most of the other PICTs (Samoa, Solomon, Papua New Guinea, Vanuatu. . .), it is less advanced [7]. If the digital transition continues with the development of equipment, the consumption of unhealthy food and drinks on these islands may also increase with longer screen times. This will probably raise public health issues in the future if the trend continues.

### Limitations

This cross-sectional study associating the factors of screen time with the consumption of unhealthy food points to the need for longitudinal follow-up studies of adolescents to further investigate the growth in digital technologies and the links with food behaviors. As this study was cross-sectional, we cannot conclude to any causal links between factors.

The small number of Polynesian adolescents in rural areas (1.85% of the sample) made it difficult to analyze this population and thus we cannot provide a reliable conclusion about this ethnic community.

Time on video game consoles was not studied in this research, which is likely why we observed no difference between girls and boys. Similarly, questions about whether young adolescents eat in front of their screens were not asked. This limits the analysis of the causal links between screen time and the consumption of unhealthy food and drink.

Concerning the food frequency questionnaire, although the study does not address obesity, it may be that social desirability bias may have led overweight adolescents to overreport or underreport food consumption [51]. One might assume that food consumption in front of the screens occurs, but the questionnaire we used could not provide such confirmation.

The screen practices of adolescents were not studied in this article. We looked at screen time to determine whether there might be an association with the consumption of unhealthy foods and drinks, without judging whether these practices were good or bad. Especially, adolescents were not asked whether they work, play, chat with relatives or do other social activities during screen time.

### Conclusion

This study revealed both a high number of digital devices available in New Caledonian households and wide disparities across urban and rural areas related to the ongoing digital development of the country. Moreover, we observed that the screen use, through time spent in front of screens, was linked to unhealthy food and drink habits. Urban Polynesian and Melanesian adolescents were especially likely to consume more unhealthy food and drinks in association with increased screen time, indicating the vulnerability of these categories of adolescents in a world of increasingly more processed foods.

We underline the importance of digital literacy skills development during adolescence combined with the monitoring of screen use by parents (at home) and teachers and educators (at school).

Future research might consider the impact of the Covid-19 pandemic and the changes in screen time in this multicultural population. This last would shed light on whether the inequalities in access to equipment between young people from rural and urban areas have been reduced or increased.

### Acknowledgments

We would like to thank the school teaching teams and administrative staffs for their help and support in our investigations, especially the Vice-Rectorat of New Caledonia. We would like to

thank Paul Zongo, Pierre-Yves Le Roux, Fabrice Wacalie, Solange Ponidjia and Emilie Paufique for their help in collecting the data.

## Author Contributions

**Writing – original draft:** Akila Nedjar-Guerre.

**Writing – review & editing:** Guillaume Wattelez, Christophe Serra-Mallol, Stéphane Frayon, Olivier Galy.

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
