## [Decision Letter · Decision Letter 0]

31 Mar 2022

PONE-D-21-27771

Adolescent screen time and unhealthy food consumption during the digital transition in New Caledonia

PLOS ONE

Dear Dr. NEDJAR-GUERRE,

Thank you for submitting your manuscript to PLOS ONE. After careful consideration, we feel that it has merit but does not fully meet PLOS ONE’s publication criteria as it currently stands. Therefore, we invite you to submit a revised version of the manuscript that addresses the points raised during the review process.

The manuscript has been evaluated by two  reviewers, and their comments are available below.

The reviewers have raised a number of concerns that need attention, and they request additional information on methodological aspects of the study and analyses.

Could you please revise the manuscript to carefully address the concerns raised?

We look forward to receiving your revised manuscript.

Kind regards,

Vanessa Carels

Staff Editor

PLOS ONE

“The author(s) received  funding from Nestlé for this work.”

“We would like to thank the school teaching teams and administrative staffs for their help and support in our investigations, especially the Vice-Rectorat of New Caledonia. We would like to thank Paul Zongo, Pierre-Yves Le Roux, Fabrice Wacalie, Solange Ponidjia and Emilie Paufique for their help in collecting the data.

The research was funded by the Nestlé Foundation and the University of New Caledonia.”

“The author(s) received  funding from Nestlé for this work.”

6. We note that you have indicated that data from this study are available upon request. PLOS only allows data to be available upon request if there are legal or ethical restrictions on sharing data publicly. For more information on unacceptable data access restrictions, please see http://journals.plos.org/plosone/s/data-availability#loc-unacceptable-data-access-restrictions.

Reviewers' comments:

Reviewer's Responses to Questions

**Comments to the Author**

1. Is the manuscript technically sound, and do the data support the conclusions?

Reviewer #1: Partly

Reviewer #2: Yes

2. Has the statistical analysis been performed appropriately and rigorously? 

Reviewer #1: Yes

Reviewer #2: Yes

3. Have the authors made all data underlying the findings in their manuscript fully available?

Reviewer #1: No

Reviewer #2: Yes

4. Is the manuscript presented in an intelligible fashion and written in standard English?

Reviewer #1: Yes

Reviewer #2: Yes

5. Review Comments to the Author

Reviewer #1: (1) If the paper is considered unsuitable for publication in its present form, does the study itself show sufficient potential that the authors should be encouraged to resubmit a revised version?

The paper is unsuitable in its current form, but given it is still relatively rare to see research by scholars of the Pacific islands, from an institution in the Pacific islands, about the Pacific islands (awesome!), it definitely merits revision and resubmission. The main points to address are outlined in the remainder of the review.

(2) What are the main claims of the paper and how significant are they for the discipline?

The paper identifies correlations between screen time and unhealthy food consumption amongst adolescents in New Caledonia. The finding itself is not particularly new or significant generally, because links between screen time and unhealthy food consumption in many parts of the world are well-established (e.g. Alex Brewis et al 2011 ‘Body norms and fat stigma’, Anne Becker 2005 ‘Television, disordered eating…’), and highlight that the content of the screen time is as important for influencing diet as the time. The same global structural inequalities underpin both; in addition, significant amounts of food marketing can occur via screens (e.g. in-game product placement, advertising, etc) and this space is largely unregulated. However, it is new to get this sort of research carried out in the islands of the Pacific by local researchers, and for this reason it should be commended.

Overall, the conclusion that monitoring of screen use is necessary lacks nuance, and it is also hard to see how this recommendation has come from the data collected. Likewise, the conclusion that dietary or health education is necessary also lacks nuance, and it is likewise hard to see how this recommendation came from the data collected. The paper would be richer engaging with some of the critiques of these to recommend new ways forward, rather than distilling recommendations to education and surveillance (the methods we have employed for the past 100 years with no really evident change - see McLennan et al 2018 'The problem with...'). Or, if these are the best ways forward, giving more evidence that they come from the data and are likely to have an impact would be useful.

(3) Are the claims properly placed in the context of the previous literature? Have the authors treated the literature fairly?

Some literature is treated really well. Other literature is not considered, and would strengthen the manuscript. In particular:

- Links between screens and unhealthy food (e.g. Maricarmen Vizcaino 2020 ‘From TVs to tablets’), and the importance of the screen content to shaping food behaviours (e.g. Brewis and Becker above)

- A critical approach to recommending health education or intervention, rather than addressing structural factors that lead to poor health, to address unhealthy food consumption in the UK (Ulijaszek and McLennan 2016 ‘Framing obesity in UK policy…’), Pacific (McLennan et al 2018, see above) or elsewhere.

- Literature on ‘screen time’ as TV time in the Pacific islands. The authors claim that the literature is sparse, but it would be useful to know what specifically has been understudied (e.g. television, computers, internet, other things?)

Some claims are a little loose and require more accuracy and/or justification and explanation throughout the paper. For example:

- “The digital transition has been accompanied by a socioeconomic transition” – what sort of socioeconomic transition does the author mean specifically, and what is their evidence for this? Or, if they just mean that “Increasing connectivity of islands to the internet has led to changes in the way people engage with screens” then say this more specifically and give evidence to back up this particular point (so, if they are claiming there has been an increase in screen use, there needs to be evidence of ‘before’ and ‘now’, not just a single data point from 2018). If the authors are going to mention things like a ‘socioeconomic transition’ they must engage with what they mean by this specifically; personally I’m not sure this particular term is needed in this paper.

- It would be useful if the authors could explain in the introduction what they mean by ‘screen time’ and/or ‘screens’. Their text implies they are talking about use of internet-connected devices, but ‘screen time’ in the past has also been used to mean television time, and there is a vast literature on this topic already. Some of the data they cite relates to the former, and some to the latter, and this needs untangling and clarification. Does ‘screen time’ mean the same thing in New Caledonia as it does in other contexts? For example, some children engaged in remote learning will spend large amounts of time at a screen, but this is very different to, say, watching commercial television or playing games (that feature advertising or not) or talking to loved ones. In this context, not all screen time may be bad – although the paper implies that this is the case. And, what is the difference between using a phone and a computer/tablet? Are there different activities? Do they serve different purposes? What about on weekends versus weekdays? What sort of screen use could be encouraged versus not? There are also a lot of positives from being connected, presumably? Or maybe not?

- It would be useful if the authors could engage critically with the notion of the ‘digital transition’. The evidence they cite indicates that growing internet connectivity has changed people’s use of technology (specifically screens, in this paper), and that the changes have been different in different contexts. This is indicative not of some sort of universal and inevitable ‘transition’ but instead context-specific changes in the ways people live, work and play. Indeed, the notion of transitions has been critiqued thoroughly in the Pacific islands context, most notably around the idea of the ‘nutrition transition’. Do they really mean increased internet connectivity of people living in the Pacific islands? If so, they could just state this rather than using the problematic notion of ‘digital transition’. The statement that New Caledonian adolescents might reach the same numbers as those in France is particularly problematic in this regard, as it reinforces colonial notions that islanders all inevitably want to ‘develop’ to become more like their colonisers.

- The authors should also be mindful that division into ‘Melanesian’ or ‘Polynesian’ is also a colonial construct – it would be useful if they could explain whether they are replicating the colonial category (and if so, why), or whether these categories are now used by the people of New Caledonia to describe themselves.

There are a number of other concepts where it would be of value if the author could engage critically with them rather than just applying Western notions to making sense of islander lives. For example:

- The notion of ‘sedentary time’ (as something to be avoided) is not necessarily useful in a Pacific islands context where sitting together is an important social practice. Not sure what the context is in NC though.

- Obesity has been demonstrated as related to much more than just diet, place of living, activity, or other individual behaviours. Indeed, the focus on individual behaviours has also been critiqued in some of the sources cited by the authors.

Finally, there is discussion about tribal activities and rural v urban; more needs to be said about these and what they mean for the research in the introduction. A map would be useful.

(4) Do the data and analyses fully support the claims? If not, what other evidence is required?

There is some confusion in the final conclusion – if I have read it correctly, the authors are right to highlight correlation between screen time and unhealthy food. However, their claim at the start of the paper – that unhealthy food consumption increases with screen time (can they show this causation?), or that health education programs are the answer, are less robust. There is plenty of literature illustrating the failure of health education programs to address the structural inequities underpinning unhealthy food consumption. There is also literature pointing out how recommending health education programs deflects blame from food companies. That the authors make this recommendation is especially problematic given the paper is funded by Nestle. Overall, this recommendation requires revision and/or considerably more justification if the paper is to be published. In addition, the authors need to critically examine the funding source, and how it can influence the sorts of research being done (or not) in New Caledonia, how it might influence research framing and conclusions, especially given that the findings are in line with global food industry lobby narratives.

The graphs do not add much value, and they seem out of place in the methodology section. I wonder whether they would be best presented in prose or even in data table format? They also require explanation in the text: What can we conclude from each figure and how does each add to the argument? A map instead would be valuable.

Data were collected prior to the COVID-19 pandemic. I think this is important to note, because the pandemic has had many effects on internet connectivity and screen use.

(5) PLOS ONE encourages authors to publish detailed protocols and algorithms as supporting information online. Do any particular methods used in the manuscript warrant such treatment? If a protocol is already provided, for example for a randomized controlled trial, are there any important deviations from it? If so, have the authors explained adequately why the deviations occurred?

N/A

(6) Are details of the methodology sufficient to allow the experiments to be reproduced?

To a certain extent yes, although bear in mind that societies change over time, so while the methods could be reproduced, it is unlikely they would yield the same results, or that the results would be interpreted in the same way.

It would be useful if the authors could add some information on how the different people listed in the acknowledgements were involved in data collection and analysis. It is a large team in the acknowledgements, and huge coverage of the nation, and this coordination in itself is an important part of the methodology because it is not easy to do this in Pacific island nations. Kudos!

The parental permission is also incredibly high - what led to this? Local connections? Reward? Another thing that would help others to reproduce some of the work should that be needed.

How did the authors go from data to the conclusions they drew?

(7) What language/s was the survey delivered in, and how might this have affected results?

Is the European standard for determining ‘urban’ appropriate and why?

Why only use responses on unhealthy foods in data analysis? This decision requires justification.

Where did students carry out the survey? Did they imagine it was school homework? Was there potential for the setting to skew or influence their responses (e.g. to impress a teacher or ensure their parents don’t find out when they use their phone?)?

It would be useful for the data collection instruments to be included as Addenda to the paper.

With whose ethics board was the study approved? Did you also need permission from schools? What were the potential harms (e.g. talking about food with adolescents can contribute to body shame or eating changes, see Brewis and Becker)?

(8) Is any software created by the authors freely available?

N/A

(9) Is the manuscript well organized and written clearly enough to be accessible to non-specialists?

Yes, overall it is well-written and easy to read, although there are a number of sweeping generalisations that are possibly too simple, and require additional nuance and/or accuracy.

(10) Is it your opinion that this manuscript contains an NIH-defined experiment of Dual Use concern?

N/A

Reviewer #2: This paper has been found of some interest in its field. Besides, the research question posed by the author is easily identifiable and recognized the original aspect. Therefore, the study design was carefully carried out.

However, only some questions emerge:

1) explain better why the author didn’t explore the time spent on video games (line 345-346);

2) why didn’t consider weight and body mass index in correlation to screen time and unhealthy food consumption? Please, add these data if available. In the other case insert this point in limitations

3) Also discuss in Discussion Section other papers conducted in area with similar cultural diversity and gradients of urbanization and digital infrastructure development (for example, Camila Wohlgemuth Schaan, et al. Unhealthy snack intake modifies the association between screen-based sedentary time and metabolic syndrome in Brazilian adolescents. Int J Behav Nutr Phys Act. 2019; 16: 115; Silvia A. González, et al. Prevalence and Associated Factors of Excessive Recreational Screen Time Among Colombian Children and Adolescents. Int J Public Health. 2022; 67: 1604217).

6. PLOS authors have the option to publish the peer review history of their article (what does this mean?). If published, this will include your full peer review and any attached files.

Reviewer #1: No

Reviewer #2: No

---

## [Author Response · Author response to Decision Letter 0]

9 Dec 2022

The answers to the reviewers are provided in the file that is uploaded to the plosone platform

---

## [Decision Letter · Decision Letter 1]

24 Apr 2023

Adolescent screen time and unhealthy food consumption in the context of digital development in New Caledonia

PONE-D-21-27771R1

Dear Dr. NEDJAR-GUERRE,

We’re pleased to inform you that your manuscript has been judged scientifically suitable for publication and will be formally accepted for publication once it meets all outstanding technical requirements.

Kind regards,

Hamid Reza Marateb, Ph.D.

Academic Editor

PLOS ONE

Additional Editor Comments (optional):

Reviewers' comments:

Reviewer's Responses to Questions

**Comments to the Author**

1. If the authors have adequately addressed your comments raised in a previous round of review and you feel that this manuscript is now acceptable for publication, you may indicate that here to bypass the “Comments to the Author” section, enter your conflict of interest statement in the “Confidential to Editor” section, and submit your "Accept" recommendation.

Reviewer #2: All comments have been addressed

Reviewer #3: All comments have been addressed

2. Is the manuscript technically sound, and do the data support the conclusions?

Reviewer #2: Yes

Reviewer #3: Yes

3. Has the statistical analysis been performed appropriately and rigorously? 

Reviewer #2: Yes

Reviewer #3: Yes

4. Have the authors made all data underlying the findings in their manuscript fully available?

Reviewer #2: Yes

Reviewer #3: No

5. Is the manuscript presented in an intelligible fashion and written in standard English?

Reviewer #2: Yes

Reviewer #3: Yes

6. Review Comments to the Author

Reviewer #2: The Authors had widely revised the paper answering to main criticisms. All the issues has been discussed. The paper has be completely rewritten.

Reviewer #3: (No Response)

7. PLOS authors have the option to publish the peer review history of their article (what does this mean?). If published, this will include your full peer review and any attached files.

Reviewer #2: No

Reviewer #3: **Yes: **Marjan Mansourian

---

## [Editor Report · Acceptance letter]

27 Apr 2023

PONE-D-21-27771R1 

Adolescent screen time and unhealthy food consumption in the context of the digital development in New Caledonia 

Dear Dr. Nedjar-Guerre:

I'm pleased to inform you that your manuscript has been deemed suitable for publication in PLOS ONE. Congratulations! Your manuscript is now with our production department. 

Kind regards, 

on behalf of

Dr. Hamid Reza Marateb 

Academic Editor

PLOS ONE